# New Drug Design Avenues Targeting Alzheimer’s Disease by Pharmacoinformatics-Aided Tools

**DOI:** 10.3390/pharmaceutics14091914

**Published:** 2022-09-09

**Authors:** Lily Arrué, Alexandra Cigna-Méndez, Tábata Barbosa, Paola Borrego-Muñoz, Silvia Struve-Villalobos, Victoria Oviedo, Claudia Martínez-García, Alexis Sepúlveda-Lara, Natalia Millán, José C. E. Márquez Montesinos, Juana Muñoz, Paula A. Santana, Carlos Peña-Varas, George E. Barreto, Janneth González, David Ramírez

**Affiliations:** 1Centro de Investigación de Estudios Avanzados del Maule (CIEAM), Vicerrectoría de Investigación y Postgrado, Universidad Católica del Maule, Talca 3480094, Chile; 2Facultad de Ingeniería, Instituto de Ciencias Químicas Aplicadas, Universidad Autónoma de Chile, Santiago 8910060, Chile; 3Departamento de Nutrición y Bioquímica, Facultad de Ciencias, Pontificia Universidad Javeriana, Bogotá 110231, Colombia; 4Escuela de Medicina, Fundación Universitaria Juan N. Corpas, Bogotá 110311, Colombia; 5Instituto de Ciencias Biomédicas, Facultad de Ciencias de la Salud, Universidad Autónoma de Chile, Temuco 4780000, Chile; 6Departamento de Farmacia, Facultad de Ciencias, Universidad Nacional de Colombia, Bogotá 111321, Colombia; 7Center for Bioinformatics, Simulation and Modeling (CBSM), Universidad de Talca, Talca 3460000, Chile; 8Departamento de Farmacología, Facultad de Ciencias Biológicas, Universidad de Concepción, Concepción 4030000, Chile; 9Department of Biological Sciences, University of Limerick, V94 T9PX Limerick, Ireland

**Keywords:** Alzheimer’s disease, drug design, computational polypharmacology, bioinformatics, pharmacoinformatics, multitarget directed ligands, protein–protein interaction network, pharmacophore

## Abstract

Neurodegenerative diseases (NDD) have been of great interest to scientists for a long time due to their multifactorial character. Among these pathologies, Alzheimer’s disease (AD) is of special relevance, and despite the existence of approved drugs for its treatment, there is still no efficient pharmacological therapy to stop, slow, or repair neurodegeneration. Existing drugs have certain disadvantages, such as lack of efficacy and side effects. Therefore, there is a real need to discover new drugs that can deal with this problem. However, as AD is multifactorial in nature with so many physiological pathways involved, the most effective approach to modulate more than one of them in a relevant manner and without undesirable consequences is through polypharmacology. In this field, there has been significant progress in recent years in terms of pharmacoinformatics tools that allow the discovery of bioactive molecules with polypharmacological profiles without the need to spend a long time and excessive resources on complex experimental designs, making the drug design and development pipeline more efficient. In this review, we present from different perspectives how pharmacoinformatics tools can be useful when drug design programs are designed to tackle complex diseases such as AD, highlighting essential concepts, showing the relevance of artificial intelligence and new trends, as well as different databases and software with their main results, emphasizing the importance of coupling wet and dry approaches in drug design and development processes.

## 1. Introduction

The World Health Organization (WHO) defines Alzheimer’s disease (AD) as a neurodegenerative disease, which is of unknown etiology characterized by cognitive impairment of memory and cognitive function [1,2], caused by a multiplicity of conditions or pathologies that lead to progressive and irreversible neurodegeneration process [3].

In this context, different hypotheses have been developed to define its physiopathological character. In recent decades, two neuropathological mechanisms have been studied and characterized in the brains of AD patients. First, the formation of amyloid plaques involving amyloid-β (Aβ) aggregation and deposition [4], and second, the formation of neurofibrillary tangles (NFTs) due to hyperphosphorylation and aggregation of tau protein [5]. In addition, the loss of connections between neurons in the brain is also involved [6] (Figure 1). Aβ and NFTs are estimated to begin about 10–20 years before cognitive function impairment manifests [7,8], in line with the most relevant hypotheses for this disease.

### 1.1. Main Hypotheses Currently Approved for AD

#### 1.1.1. Cholinergic Hypothesis of AD

This is the first hypothesis, which suggests that a dysfunction of the cholinergic neurons, which contributes significantly to the cognitive failure observed in AD, is the cause of the disease. However, some studies also claim that acetylcholinesterase (AChE) activity is up-regulated or unaffected in patients with mild cognitive impairment or early AD, leading to question the validity of this hypothesis [9].

#### 1.1.2. Amyloid Hypothesis of AD

Aβ peptide is produced by abnormal cleavage of amyloid precursor protein (APP) (a transmembrane glycoprotein expressed in different cells, including neurons and glia). Under normal physiological conditions, the cleaved products of APP are soluble peptides easily eliminated by the body. However, in AD, APP is cleaved by α-secretase, β-secretase, and γ-secretase, generating insoluble peptides which are then secreted by neurons into the extracellular space where senile plaques are formed due to accumulation, oligomerization, and deposition of the Aβ peptide, promoting a neuroinflammatory process and generating structural damages [10]. Although this hypothesis has been supported for several years, there is controversy regarding the details of amyloid pathology and its relationship with AD because some patients have been found with amyloid pathology but without AD symptoms [11]. Along with this, the different oligomers produced by the deficient activity of the secretase have been investigated, where some of the different sizes, in different concentrations, and with different outcomes in patients have been discovered. Initially, it was believed that beta-amyloid consisted of a maximum of 42 amino acids [12], but in 2006 Lesné et al., had published an article reporting the finding of an Aβ*56 oligomers, which would have direct consequences on the memory function of test mice [13]. Despite the efforts of different scientists to elucidate this point, some of them have not been able to reproduce results on this oligomer, and others have found contrasting answers [14]. Research remains to be performed on the incidences of the length of the oligomer, its solubility characteristic, its concentration in different organisms depending on the pathology, and its relationship with Alzheimer’s disease to be able to restructure this hypothesis in an effective way.

#### 1.1.3. Tau Hypothesis of AD

Tau protein is found mainly in axons. Its function relates to the binding of microtubules to stabilize the neuronal cytoskeleton. Under AD, the tau hyperphosphorylation form decreases its affinity for microtubules, causing it to aggregate in an insoluble form (known as paired helical filaments), resulting in the transformation of tau into NFTs. In the formation of NFTs, the hyperphosphorylated tau disrupt microtubule function, which leads to impaired axonal transport, blocking the transport of nutrients and essential molecules within neurons [15].

### 1.2. Other Hypotheses and New State of the Art in Pathophysiology of AD

In addition to the pathophysiological hypotheses that have recently been accepted to explain the mechanisms of AD, there are several hypotheses that have gained different weight depending on the studies they are presented. For example, hypertension and cardiovascular diseases have been related to AD, giving rise to the angiotensin and vascular hypothesis, respectively. The former describes the relationship between the renin–angiotensin system, whose components are altered in AD due to its important role in blood pressure and cardiovascular regulation [16]. The second relates to the lack of blood flow to the brain with vascular dementia and consequent cell death, where cellular aging and brain trauma could be risk factors for AD [17]. On the other hand, hypotheses such as oxidative stress [18] explain how abnormalities in neuronal cells would increase under this condition, which leads in many of these cases to some apoptotic mechanisms associated with cognitive dysfunction and dementia. Moreover, some hypotheses are more related to environmental and lifestyle factors, such as the mercury hypothesis [19], which explains that mercury contamination of the body is capable of causing many of the various alterations present in Alzheimer’s patients. Likewise, the increase in cholesterol at the brain level [20] and the lack of vitamin D [21] are two hypotheses that would explain the dysfunctionality of the organism that is characteristic of AD. In the case of vitamin D, its alterations can mimic the amyloid pathways, which could also be affected by alterations in calcium levels [22], forming other hypotheses associated with this complex disease. In recent literature, there are also more specific hypotheses, such as the hypothesis of mitochondria-associated endoplasmic reticulum membranes [23] and how the functions located in this part of the cell increase their expression in Alzheimer’s patients. There are also some related to exposure to microorganisms of more biological origin or the viral hypothesis that relates the presence of common viruses such as herpes to the development of the other pathological hypotheses of AD [24].

All this evidence allows us to observe the complex nature of AD. There has been a limited success (almost none) in drug development and repurposing efforts for its effective treatment, in part due to the fact that the drug design process targeting AD has been flanked by a reductionist “1-target <–> 1-drug” model. So far, this approach has not provided an effective therapeutic alternative, possibly due to the multifactorial nature of this pathology. Therefore, there is a need to shift the focus of current AD drug discovery research towards designing better therapeutic solutions for simultaneously targeting the multiple pathological mechanisms responsible for the initiation and advancement of AD. Regarding the efforts made by the scientific community in recent years to deal with this challenge and this multifactorial disease, either from clinical treatments, drug design, and laboratory tests (among others), it is important to note that currently, about 2880 Alzheimer-related projects are being developed, according to the clinical trials database. As shown in Figure 2, only half of them have been completed, 5% are active, and approximately 19% are in the start-up phase. This shows the high interest there is currently in conducting clinical research in this area. In these projects, several countries have taken the lead, including the United States (1479), France (245), China (89), Canada (86), Korea (68), the United Kingdom (67), Spain (66), and Germany (55) (Data retrieved on July 2022 from www.clinicaltrials.gov).

Within the complex research that aims to face this challenge, drug design by pharmacoinformatics-aided tools has taken on significant relevance. Currently, several strategies, methods, and tools have been developed to progress in this field, which can be found in different reviews where their scopes are specified [25,26]. In this review, three strategies to improve the pharmacological AD space are highlighted because of their relevance in drug design as well as AD. First, Multi Target Directed Ligands (MTDLs) approach focuses on designing drugs that simultaneously hit several relevant targets [27]. Second, drug repurposing, which refers to revealing new uses (or purposes) for prescribed ligands or drugs [28], and the third strategy is called systems pharmacology, which takes advantage of the huge amount of biological data, available mathematical and pharmacology models to propose or predict an optimal therapy with the integrative vision of the systems approach. An analytical description of the advantages and limitations is described elsewhere [29,30].

These approaches use several computational techniques to aim their purposes, such as molecular docking to predict the best conformational stage of a small molecule into a macromolecule [31,32], together with virtual screening, used to predict putative bioactive ligands from large databases of small molecules [33,34,35], and pharmacophore modeling, which represents a three-dimensional ensemble of chemically defined interactions of a compound with its possible receptor [36,37]. This concept is commonly applied to perform ADME-Tox prediction, side effects modeling, off-target prediction, target identifications, and virtual screening campaigns. Finally, molecular dynamics simulation is one of the most computationally expensive and time-consuming classical techniques commonly applied in computer-aided drug design (CADD). It uses the structural information available for the receptor and its ligand while employing a forcefield to predict the evolution of the system along a given simulation time [38,39]. Additionally, due to the relevance of Alzheimer’s disease within neurodegenerative diseases (NDD) and drug design, different databases and software have been developed exclusively for the study of Alzheimer’s disease; some examples of these are listed in Table 1.

## 2. Advances Achieved by Bioinformatics Tools in the Diagnosis of AD

One of the main difficulties in the treatment of this disease is that an effective way to diagnose it in the early stages of the patient’s life has not yet been determined. On the contrary, it is only possible to confirm the development of the disease post-mortem by detecting some of the pathophysiological characteristics of AD in the brains of patients [44]. The key steps in the diagnosis of AD include consideration of the patient’s clinical history, physical examination, neuropsychological testing, and neuroimaging. It has been studied and concluded by L. Guzman-Martinez et al. that there are genes that confer susceptibility to this disease and that they are related to lifestyle, being healthy living such as exercise, balanced diet, and constant brain activity a way to mitigate the probabilities of suffering AD [45]. Along with this, it was determined that prolonged confinement in both young people and adults could increase the risk of suffering from this disease precisely because of the absence of the key aspects of a healthy lifestyle [45].

A common practice in the diagnosis of AD is the use of Magnetic Resonance Imaging (MRI). The use of these images represents a challenge, given to high similarities across stages of dementia in AD brains, as well as with healthy brains of older people [2]. In this sense, a way to refine this is through bioinformatics, which is being used to achieve a more accurate AD diagnosis using MRI images without the intervention of the human eye, leading to early diagnosis and better therapeutic efficacies. This improvement has been accomplished by implementing deep learning (DL) technologies to identify changes that occur in the brain even before the first symptoms arrive, such as the change of the early medial temporal lobe, a pattern characteristic of AD caused by both amyloid-β plaques, and neurofibrillary tangles accumulation [46]. Nevertheless, bioinformatics is not only being used for AD diagnosis based on image analysis; the screening and identification of key genes using gene set enrichment analysis (GSEA) are also being examined for early AD diagnosis in clinical practice [47]. Based on functional analyses, differentially expressed genes and microRNAs have been proposed to be closely related to AD through a comparison against pathology databases such as the Comparative Toxicogenomics Database (CTD) [48], which integrate large amounts of data about AD and promote the discovery of new biomarkers. Some other AD databases are hu.MAP [49], ADNI [50], NIAGADS [43], and HENA [42].

On the other hand, clinical biomarkers such as Aβ42, T-tau, and P-tau proteins are extracted from cerebrospinal fluid to detect the pathophysiology of AD. However, this method is too invasive and painful for the patient. Therefore, there is a great effort to discover biomarkers for AD diagnosis that can be detected using less invasive methods [51]. In this sense, techniques on different molecular levels, such as genomics (DNA), transcriptomics (mRNA and non-coding RNAs), proteomics (proteins), and metabolomics (metabolites), are being implemented to identify the pathways that lead to neuronal death and the biomolecular markers associated with the AD neuropathology [52].

Omics analysis approaches are mainly focused on statistical analysis or machine learning (ML) analysis. ML analysis integrates complex datasets derived from omics techniques in scenarios using supervised (e.g., Random Forest and Artificial Neural Network) or unsupervised algorithms (e.g., Hierarchical Clustering, k-means Clustering, and Artificial Neural Network), where traditional statistical methods are insufficient [52]. The use of omics has given rise to a large number of molecular profiles and datasets, which are crucial not only to an early diagnosis but also for understanding the complexity of AD and, eventually, for creating a personalized treatment using precision medicine [52].

An irrefutable fact is that, by the time AD is detected and diagnosed, it is probably too late to cure the patient because there is a high neurodegeneration level. For that reason, therapies are also focused on the early detection of AD [53,54,55]. Although NFTs and Aβ in the brain represent major hallmarks of NDD, therapies aiming to reduce the amyloid have not shown any breakthrough in reversing mild cognitive impairment [3]. This background allows us to understand that there is no clear and complete statement regarding the causality of AD [56]. Therefore, in addition to studying this aspect in depth, it is indispensable to have a method for early diagnosis and novel therapy. Clearly, out-of-the-box alternatives to obtain therapeutic results from a system-pharmacology drug design perspective must be sought using data generated over the decades.

## 3. Current Therapeutic Strategies against AD

The treatment of this disease has been approached from different points, from unconventional investigations, such as alternative therapies, to classic pharmacological ones. Table 2 shows a summary and representative description of the type of treatments that have been carried out in the last two decades to improve the condition of this disease and try to control it.

Considering that drugs are the most widely used method in the treatments described above and that most of the approved drugs are small molecules, it is necessary to mention that there are six drugs that have been approved by the U.S. Food and Drug Administration (FDA) for the treatment of AD. These include donepezil, rivastigmine, and galantamine, which are acetylcholinesterase inhibitors (AChEIs) preferentially used during the early mild and moderate phases of the disease [57], and whose chemical structures are shown in Figure 3. Unfortunately, these drugs only temporarily alleviate cognitive symptoms without having an effect on the progression of this disease [58]. In advanced stages, cholinesterase inhibitors are often combined with memantine, a non-competitive N-methyl-D-aspartate (NMDA) receptor antagonist [59,60]. Another relevant cholinesterase inhibitor is tacrine, the first drug approved for AD [61], which has been withdrawn from treatments [62] because although it generates cognitive improvements as a palliative benefit, it was discovered at the clinical level that it could cause hepatotoxicity [63]. This controversy of having a structure that generates the desired improvements but also has strong side effects gave rise to research into new tacrine-based derivatives that could solve this problem [64,65,66,67].

Although these drugs cannot delay neurodegenerative progression, they temporarily improve the cognitive function of cholinergic and glutamatergic neurotransmission [69], improving the patient’s quality of life in a palliative way. However, they have the inconvenience of presenting some side effects such as gastrointestinal complications [70], muscle problems in anesthetized patients, slow heartbeat and fainting, as well as seizures [71].

Recently (June 2021), the FDA approved Biogen’s drug aducanumab (Aduhelm^TM^), a disease-modifying monoclonal antibody that, upon entry into the brain, interacts with parenchymal amyloid and decreases the concentration of Aβ in a dose-dependent manner [72]. In the same year, the FDA limited its approval only to patients with mild cognitive impairment or mild dementia due to AD [73]. Since its fast approval in 2021, it has been controversial in the health care field and the scientific community [74]. However, the approval of aducanumab has paved the way for more extensive and reliable development of monoclonal antibodies to modulate multiple AD targets in the future.

Over the last decade, some drug delivery strategies such as nanoparticles have been proposed, which have become useful for blood–brain barrier (BBB) transport, turning into an important approach to overcome the side effects problem, reducing the impact of these drugs on the peripheral level. In AD, it has been shown that by using nano-based drug delivery, it has been possible to decrease Aβ production, aggregation, and clearance, as well as tau phosphorylation and packaging [75], in which significant progress has been demonstrated [76]. Despite all efforts, it has not yet been possible to identify the reason that clinical trials against AD continue to fail [77]. However, it is becoming clear from research in the last decades that the use of polypharmacological therapies could be a starting point to deal with the multifactorial nature of complex diseases such as AD. Regarding the number of molecules currently under development, both small molecules and antibodies, there are about 200 in clinical trials that could offer some hope for the future (data retrieved in July 2022 from the ChEMBL database). Figure 4 shows a graphical summary of the types of molecules being studied and the phase in which each one is found.

## 4. Computational Polypharmacology Applied to Multitarget Drug Design in AD

Polypharmacology is defined as the design or use of pharmaceutical agents that act on multiple targets or disease pathways [78]. Recent research in pharmacology has changed the paradigm of drug discovery for complex and multifactorial diseases such as cancer, mood disorders, and NDD, given that these diseases result from a complex network of molecular events not based on a single target. Although this search has been commonly undertaken from a ligand synthesis perspective, computational approaches have lately become consolidated in multitarget drug discovery [79]. These methods stem from the 2D or 3D shape and chemical similarity evaluation, target and binding site similarity assessment, graph theory and modeling, docking methods, pharmacophore analyses, machine-learning algorithms, and chemogenomics [79], offering a complete approach to study the binding mechanisms and interactions of certain molecules to the targets to be modulated. Therefore, several authors have worked on the use of computational polypharmacology methods for the design and development of drugs for the treatment of Alzheimer’s disease.

In 2020, Oddson et al. [80] developed a high-performance virtual screening (HTVS) to identify new modulators of two targets involved in AD: AChE and alpha-7 nicotinic acetylcholine receptors (nAChR α7), confirming that the HTVS approach can be applied in the search for new drugs with dual activity. Similar to the research of Montanari [57], where they also performed molecular docking and molecular dynamics to study the multitarget behavior of a series of coumarin-based derivatives. Another research performed in 2021 studied about 134 secondary metabolites of *Gongronema latifolium* leaves using HTVS against protein kinases LRRK2, GSK3β, and MAPK14, which have been associated with the onset of Parkinson’s and Alzheimer’s disease [81], providing a complete analysis through different computational tools that are of great use for polypharmacology. Following the same line, recently, Nozal et al. combined fragments that inhibit key protein kinases involved in the main molecular pathophysiology pathways of AD, such as tau aggregation, neuroinflammation, and decreased neurogenesis, and developed novel MTDLs with the capability to inhibit LRRK2, CK1δ, and GSKβ kinases as well as BACE1. They reported well-balanced MTDLs with in vitro activity in three different relevant targets and efficacy in two cellular models of AD. Furthermore, computational studies confirmed how these compounds adequately accommodate into the long and rather narrow BACE1 catalytic site. Finally, they employed in situ click chemistry using BACE1 as protein template as a versatile synthetic tool that allowed us to obtain further MTDLs [82].

These types of findings show how computational polypharmacology (always coupled with experimental validation) can contribute to a thorough understanding of the binding mode of ligands at their binding site, helping to reveal indispensable details for the proper design of MTDLs.

### Multi-Target Directed Ligands (MTDLs) for AD

MTDLs can simultaneously modulate two or more targets, implying that these targets may have structural or electrostatic similarities—common pharmacophoric features—which enables them to be modulated by the same chemical entity [83]. There is currently much interest in the development of multitarget drugs for AD. As shown in Figure 5, more than 700,000 active compounds are associated with more than 2000 targets involved in the disease, where a large percentage of them are able to modulate two or more targets simultaneously. Further information was deposited in the Open Science Framework project “New drug design avenues targeting Alzheimer’s disease by pharmacoinformatics-aided tools” (https://osf.io/by86r/).

On the other hand, similarities between interaction and binding sites and their main characteristics for ligands and specific target–ligand complexes should be taken into account when multitarget drug design campaigns are being implemented, as reported by Nuñez-Vivanco et al. [84] for dopamine and/or serotonin transporters and MAO enzymes through polypharmacology tools. 

The ligand-based drug design (LBDD) approach is often used to outline novel MTDLs [85], and it is possible to design hybrids based on different ligands by observing their structure. In the MTDLs design pipeline, there are three possible ways to conjugate the desired parts by linking, fusing, or merging, which will result in a hybrid molecule with modulated properties according to the proposed structure and bonding method, as shown in Figure 6. A well-known case relates to the tacrine hybrids, where different bioactive compounds were obtained by using tacrine (an MTDL with activity against AChE–IC_50_ = 0.42 µM and BuChE–IC_50_ = 45.8 µM [65]) as the principal scaffold. For instance, tacrine–melatonin and tacrine–hydroxyquinoline hybrids present antioxidant properties and maintain cholinesterase inhibitory activity [86]. Tacrine–flavonoid hybrids have shown a very prominent inhibitory activity against BACE1 and AChE (low pM range), which are ~10,000-fold more potent than the tacrine precursor [87]. Novel hybrids in this category have been designed and synthesized by the covalent linking of tacrine and the Aβ aggregation inhibitor dipicolylamine. The products are dimers with a potent inhibitory effect on AChE and Aβ, decreasing tau phosphorylation, preventing synaptic toxicity, and inhibiting neuroinflammation [88].

MTDLs developed by structure-based drug design (SBDD) methods [90,91] have proven to be successful due to the increasing availability of structural data for key targets in AD (crystallographic, NMR, and CryoEM structures). SBDD methods have led to understanding the importance, for example, of pi-interactions to inhibit AChE in the anionic catalytic site (orthosteric binding site) and the peripheral anionic site (allosteric binding site). These approaches also led to the identification of key residues at the BACE1 catalytic site. Dominguez et al., reported, the MTDL compound 3f with activity against AChE (IC_50_ = 14 µM), BuChE (IC_50_ = 7.1 µM), and BACE1 (IC_50_ = 3.1 µM), as well as the capability to inhibit Aβ peptide (28% at 100 µM) [91]. 

Due to the growing interest in finding structural similarities among key targets, several computational polypharmacological tools, such as Geomfinder [92], 3D-PP [93], ProBiS [94], ProCare [95], PocketMatch [96], and other tools and protocols have been described and discussed in the literature to tackle the binding site (BS) comparison problem [97,98,99,100]. The efficiency proven by these methods demonstrates the need to address the study of common characteristics among both ligands and targets to establish a rational design of MTDLs through computational polypharmacology. Some examples of compounds designed according to these strategies, and which have shown multitarget activity in AD are shown in Table 3.

## 5. Pharmacoinformatics Tools in Drug Design against AD

The application of well-known pharmacological models to study how different targets are modulated, together with the tools provided by medicinal chemistry, is fundamental to exploring the chemical space of new bioactive compounds. Furthermore, the use of software and servers such as ligand and protein databases allow for simulating the interaction between drug–protein and/or protein–protein, in which it will be possible to perform the analysis of networks and the implementation of ML models. When framed in the paradigm of systems pharmacology, these methods are applied to address the different phases of the drug discovery process against AD (preclinical and clinical phases) and could be the key to overcoming the current low success rate when designing drugs against this complex and devastating pathology. In recent years, pharmacoinformatics tools have been used to enhance drug design processes, i.e., to identify new targets in AD [107] and neurodegenerative dementias [108], to study traditional Chinese medicine in AD [109], to discover new MTDLs for AD by analyzing ligand-protein interaction networks [110], and to explore new mechanistic insights into AD through protein–protein interaction networks (PPIs) [111]. Pharmacoinformatics has undergone exponential growth, changing the way drug research and design are carried out. Currently, different servers used for pharmacoinformatics purposes have been built to support the discovery of novel therapeutic alternatives against AD.

Finally, all these tools, made up of databases, software, and different analysis methods, have been growing along with technology and scientific innovation, allowing the optimization of resources and accurate data validation, among other things. In this sense, some strategies, such as pharmacophore modeling and the use of artificial intelligence, are crucial stones in the construction of these new trends.

### 5.1. New Opportunities in Drug Discovery—Pharmacophore Modeling

Macromolecular structures (such as proteins) bind to small organic molecules, where they can trigger functional modulations and, thus, biological responses. The union of their ligands with their macromolecular targets is mainly based on the set of chemical interactions, such as hydrogen bonds, ionic, or lipophilic contacts. Thus, 3D pharmacophores represent an intuitive and powerful description of these interaction patterns [112]. The official IUPAC definition for this term describes pharmacophores as “the ensemble of steric and electronic features that is necessary to ensure the optimal supra-molecular interactions with a specific biological target structure and to trigger (or to block) its biological response” [113,114].

Moreover, these pharmacophores are not a particular set of functional groups or structural fragments, instead are an abstract description of physicochemical, steric, and electronic characteristics describing properties of molecules that are indispensable for energetically favorable ligand-target interactions (pharmacophore features), such as hydrophobic areas, aromatic rings, hydrogen bond acceptors and donors, as well as ionizable groups [115]. If the molecules possess similar pharmacophoric patterns, these can therefore be assumed to be recognized by the same binding site of a given biological target and thus also show similar pharmacological profiles [116].

#### Pharmacophore Modeling Classification

Pharmacophore generation can be performed by obtaining information from ligands, from the receptor without ligand (apo form), or from interactions described in receptor–ligand complexes, as shown in Figure 7 and as explained next. *Ligand-based modeling:* Usually, the pharmacophore builder algorithms first perform steps where quick distance checking takes place. Then, a 3D alignment of different active compounds and their conformations is computed to compare the location of the pharmacophoric features [117,118]. *Apo-based modeling:* Molecular field-based methods could accomplish the labor of apo-pharmacophore modeling. First, a grid will be placed in the putative and predefined binding site. Then, this space will be sampled by several probes to explore target-probe interactions, miming the interaction of ligand functional groups and their target. Next, an energy calculation will take place between probes and the atoms from the cavity to identify favorable interactions. Finally, the local minimum of those calculations will be translated, such as pharmacophore features [119,120,121]. *Complex-based modeling:* Macromolecule–ligand systems could be available via RMN, X-ray, and/or CryoEM, as well as molecular modeling solutions. A set of previously defined chemical and geometrics criteria will be identified and grouped into pharmacophore features [122]. Likewise, complex-based methods could fulfill this assignment, employing the previously mentioned molecular strategies [123].

### 5.2. Machine Learning and Artificial Intelligence to Enhance Drug Design against AD

Artificial Intelligence (AI) offers a wide variety of methods to analyze large and complex data in order to improve the understanding of different diseases, especially useful in the case of complex diseases such as AD. Among the most used AI methods is ML, consisting of a collection of data analysis techniques that aim to generate predictive models for classification, regression, and clustering. Another widely used AI method is DL which uses algorithms that can learn relationships between inputs and outputs by modeling highly non-linear interactions in higher representations at a more abstract level [124].

AD research using ML continues to evolve, improving performance by incorporating additional hybrid data types such as omics data and increasing transparency with explainable approaches that add insights into specific features and mechanisms related to the disease. AI has also been used to prioritize or infer repositionable drugs for AD, using DL extracting low-dimensional representations of a high-dimensional protein–protein interaction network to infer potential drug target genes [125], and even ML has been used to identify candidates for AD drug repurposing [126]. This offers a great opportunity for drug discovery and development, as ML approaches offer a set of tools that can improve decision-making for well-specified questions with abundant, high-quality data, thereby optimizing the development of new drugs [127]. It is also worth noting that ML methods are highly data-driven, and high-quality datasets are required to build suitable models. For example, data derived from databases such as PubChem and ChEMBL offers complete data related to bioactive ligands and their targets, indications, clinical phases, etc. Other databases, such as the Open Target Platform [128] or the Therapeutic Target Database [129], provide valuable information about known and explored therapeutic targets, the targeted disease, pathway information, and the corresponding drugs directed at each of these targets. Other datasets such as OASIS [130,131] and ADNI [132,133,134,135] had been generated to perform, for instance, early-stage AD prediction using ML models.

As an example of the usefulness of these tools, some studies have used random forest and support vector machines ML algorithms, which is a type of supervised learning [136] as the primary method for screening gamma-secretase inhibitors (675 inhibitory and 758 non-inhibitory compounds) using 3D structures to calculate 189 molecular descriptors, including constitutional, quantum chemical, topological and geometric descriptors. The results included 368 possible gamma-secretase inhibitors [137]. Another study developed a Bayesian ML model based on data available from ChEMBL and PubChem of AD-related proteins, where they sought to identify a new small molecule that could be administered as a treatment for AD, finding GSK3β (the protein that phosphorylates the tau protein) as a target of interest [138].

## 6. Drug Repurposing Strategies

One of the strategies used to face the challenge of designing drugs quickly, safely, and efficiently is drug repositioning. This strategy consists of assigning new indications for drugs that already exist and are used in some described pathology. Its advantages include knowledge of the drug, progress in clinical trials, and, therefore, management and understanding of its pharmacokinetics and the effects it may cause, according to its previous use [139]. As a very close example, for AD, there is the use of galantamine, one of the drugs approved by the FDA to treat this disease, which has its origin in the treatment of poliomyelitis and was repositioned for AD treatment. Similarly, fluoxetine and levetiracetam, among others, which have serotonin reuptake inhibitors and antiepileptic functions, respectively, have shown significant results in the treatment of AD [140]. In order to develop this strategy and perform an exhaustive search among the wide number of drugs that currently exist and their respective reported purposes or targets, pharmacoinformatics tools play a crucial role. For example, using network pharmacology and analyzing data from the ChEMBL database, a drug–protein interaction network (DPI), referring to proteins and drugs in AD, was built. It is possible to identify three multitarget approved drugs (rivastigmine, memantine, and donepezil), as well as five single-target approved drugs (aducanumab, florbetapir, galantamine, florbetaben, and flumetamol) currently indicated to treat AD (Figure 8). In addition, several FDA-approved drugs present activity against one or more AD targets, which makes them potential candidates for drug repurposing against AD. More detailed information about single and multiple drugs and AD targets can be found in Appendix A.

Drug repurposing has not only been used in NDD, but in general for complex diseases such as diabetes, psychosis, or cancer. An example of the latter is the drug Raltegravir as a possible complementary drug therapy, which is initially used as an HIV-1 integrase inhibitor [141], or the drug repurposing campaigns to treat COVID-19 [142], which, in view of the global pandemic, requires rapid and advanced solutions.

Another advantage of this strategy is that in conjunction with the use of PPIs, DL, and ML tools, it is possible to establish a relationship between known and potential drugs for a given treatment, based on their structure, and also to relate the targets involved, which allows a key clue in the drug design challenge [143]. For example, using the comparison of genes through pharmacoinformatics tools, it was possible to suggest new drugs interacting with targets associated with NDD, such as AD. A study shows that through this technique, 27 drugs were identified [144], showing potential activity against AD, opening doors to new challenges and approaches in the development of new treatments.

## 7. Applications of System Pharmacology in Drug Design against AD

The contributions of system pharmacology in drug design are recent and numerous, highlighting the need to understand that a single target does not describe the entire physiopathology of a given disease and that a single drug will not provide a final solution, revealing that the relationship between them opens doors not only to the design but also to the association of symptoms from the clinical side. An excellent way to address this point is through systems pharmacology and the use of networks and data mining, as shown in Figure 9. These methods can also be used to predict new drug targets based on the relationship between their functions and the reported interaction profiles with known ligands, allowing the integration of this information into interaction maps that give a more comprehensive view of the key components of the pathology. This complex model also allows the characterization of common pharmacophore features among ligands and related targets, and thus possibly involved in pathology pathways. From here, it would be possible to work in numerous ways. For instance, with subsequent virtual screening (based on ligands, structures, and pharmacophores) in order to predict new potential protein–ligand interactions (PLI) and identify active or inactive elements and scaffolds, among others. In this way, it can be explored the interaction and relationship between the key factors of the disease. Moreover, it may be possible to discover some components that have been seen as secondary but could play a relevant role, making it easier to face the multifactorial challenge of the disease from a more systematic point of view. In particular, these are necessary tools for the treatment of multifactorial diseases such as AD and NDD in general [51].

The systems pharmacology approach is a useful perspective to understand the molecular mechanisms involved in a given pathology and also to postulate new targets and predict the response of existing drugs and their adverse effects [145], being a way of tackling diseases such as AD, where it is necessary to consider the multifactorial nature in an integrative manner.

Systems pharmacology is based on the integration of data from the omics sciences, in order to understand the activity of drugs in vivo at the molecular, tissue, organismic, and cellular levels [146]. To do this, it integrates models based on pharmacokinetics-pharmacodynamics and disease systems [147], giving way to the development of predictive and quantitative interaction network modeling that allows explaining the adverse effects of drugs [148] through the understanding and graphic vision of the relationship between the different targets involved. In addition, given the ability to integrate data in a massive way, it is possible to use information reported in recent years to obtain new approaches in clinical therapy. As shown by Nguyen et al. [149], they identified targets related to metabolism and memory, such as bradykinin receptor 2 and DLG4 receptor involved in memory and cognition, which showed to be involved in a network of dementia-associated targets.

This type of network can help both to identify the consequences of modulating one target or another, or what type of signaling should be targeted, and even to give an idea of drugs that can regulate as inhibitors, agonists, or antagonists, at the principal or allosteric binding sites, depending on the relevance of the target for the study and the network that surrounds it.

The use of genomic data represents a great advance in precision therapy since it considers the possible polymorphisms present in the genome, which could cause variations in response to the drug [150]. The incorporation of this field in pharmacology has allowed us not only to understand the physiology of the disease and its response to medications [148] but has also given way to the characterization of interactions within the biological network and its influence on the identification of new therapeutic targets and the discovery, development, and repositioning of drugs [150].

## 8. Challenges and Future Perspectives 

In order to estimate the future and projections of research in this area, it is important to evaluate how it has evolved over the years. As shown in Figure 10, it is a research area that, year after year, presents more and more associated projects, demonstrating a high growth and interest by the scientific community. In addition, it is known that for the year 2023, there are already 10 proposed projects beginning to be recruited. Now the challenge remains to understand and approach this disease from more complex perspectives that allow uniting the efforts already made to find a more concrete solution or to build it with a solid base.

Understanding the need to design drugs as MTDLs, capable of simultaneously modulating several targets in the effort to reduce or reverse the pathological manifestations of a multifactorial disease, might be the cornerstone of the challenge in drug design for NDD and AD. Once this is in mind, the use of tools that allows quickness and efficiency, together with cost savings through predictive simulation and data preprocessing, are promising aspects in facing pathologies such as AD. The contributions of computational polypharmacology and pharmacoinformatics to the design of drugs for multifactorial diseases respond to these needs as a key part of the research in recent years. The current availability of various pharmacoinformatics tools, the use of networks and exhaustive analysis through artificial intelligence, and the availability of constructs such as pharmacophores to direct this process are certain fundamental elements for the development of the coming decades. From now on, knowing the different tools and software available for drug design, the challenge remains to apply them efficiently and to continue advancing in this field.

## Figures and Tables

**Figure 1 pharmaceutics-14-01914-f001:**
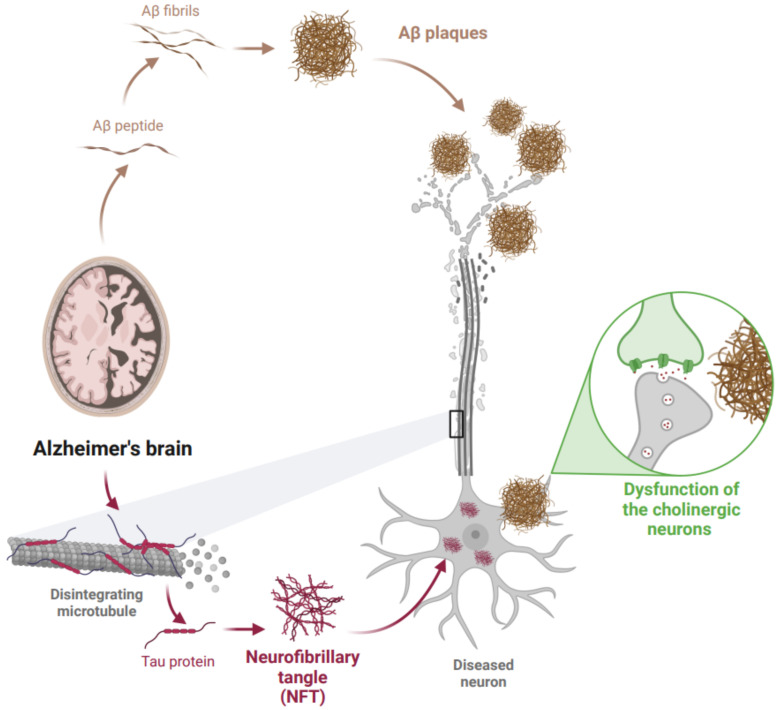
Pathophysiology of the main hypotheses of Alzheimer’s disease. Created with BioRender.com.

**Figure 2 pharmaceutics-14-01914-f002:**
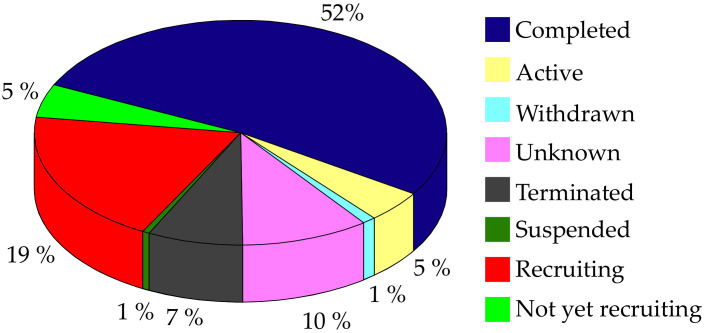
Percentage and status of AD-related projects that have been carried out in the last two decades according to the ClinicalTrials.gov database. Data retrieved on July 2022.

**Figure 3 pharmaceutics-14-01914-f003:**
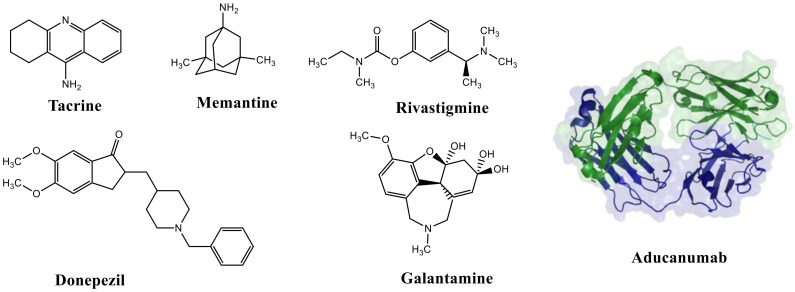
FDA-approved drugs for the treatment of Alzheimer’s disease. Small molecules (tacrine, memantine, rivastigmine, donepezil, and galantamine) and the monoclonal antibody aducanumab (where chain A is shown in green, chain B in blue and chain C in cyan.) PDB code: 6CO3 [68].

**Figure 4 pharmaceutics-14-01914-f004:**
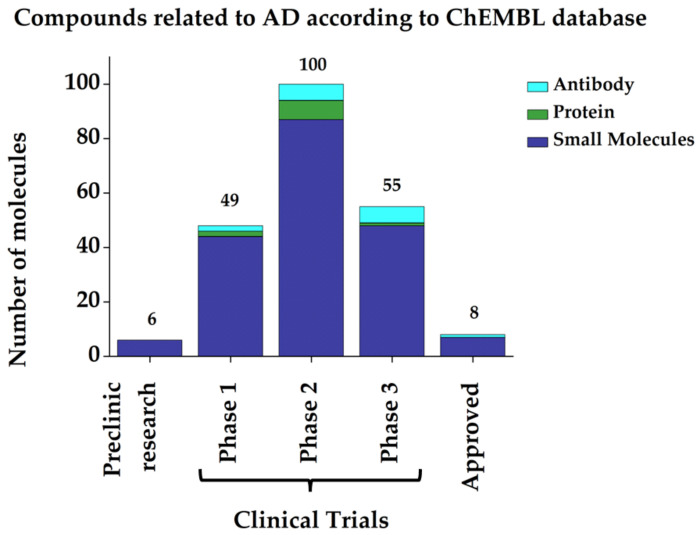
Graphical representation of the number and status of molecules related to AD. (Data retrieved in July 2022 from ChEMBL database).

**Figure 5 pharmaceutics-14-01914-f005:**
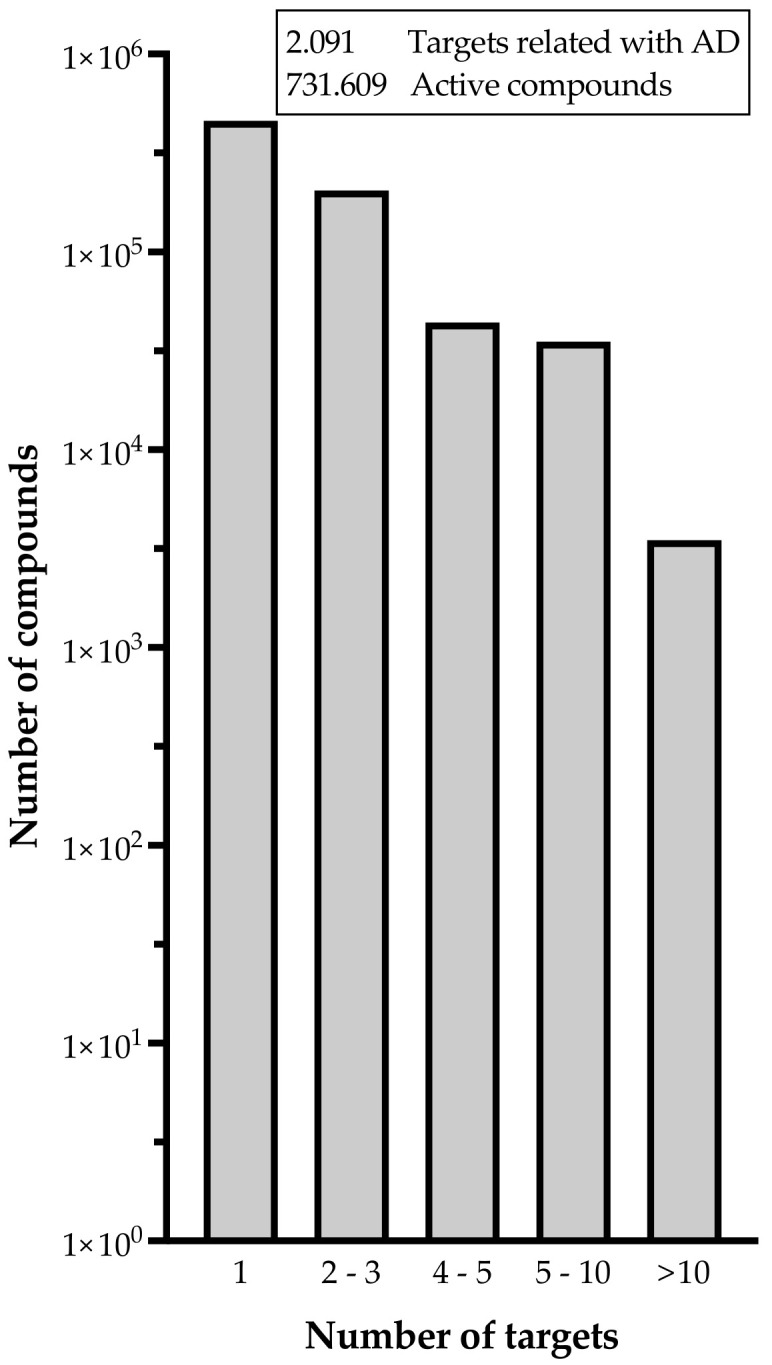
Number of active compounds related to Alzheimer’s disease and the respective number of targets they can modulate simultaneously (Data retrieved in August 2022 from ChEMBL database v31). To process the data, first targets related to AD were extracted from the Open Target Platform, then data was enriched with Uniprot IDs via Uniprot API and a local mirror of ChEMBL database v31. For those proteins with bioactive reports or reported as part of drug mechanisms on ChEMBL, we retrieved the targets and drugs names, compounds ChEMBL IDs, pchembl_value related to the activity, and the drug or compound phases of developments as itself and for the indication (Alzheimer in this case). Uniprot API queries, local ChEMBL queries, and data handling were performed with KNIME 4.6.1 platform. Detailed information can be found at https://osf.io/by86r/.

**Figure 6 pharmaceutics-14-01914-f006:**
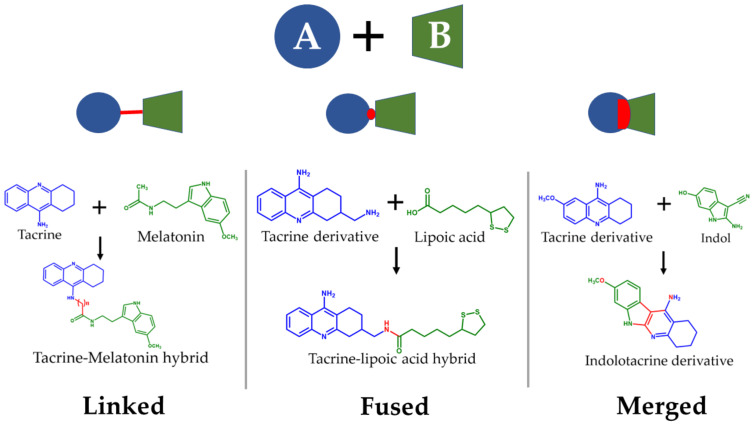
Possible strategies to design MTDLs for AD: Linked [86], fused [66], and merged [89].

**Figure 7 pharmaceutics-14-01914-f007:**
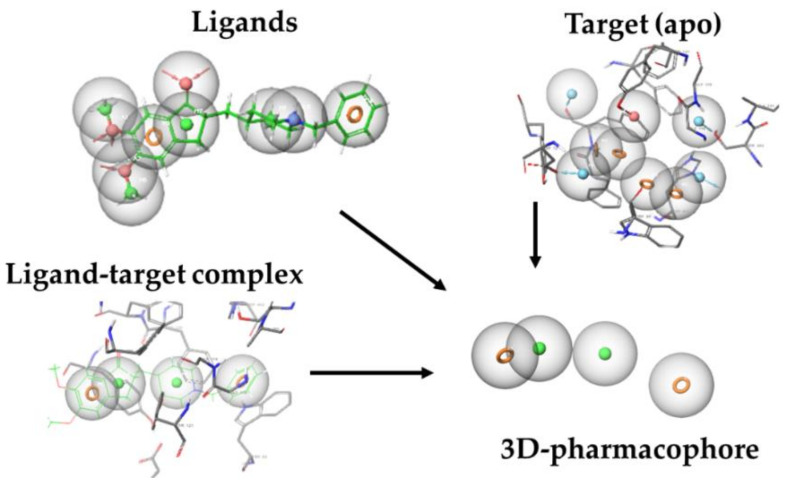
Example of pharmacophore generation for acetylcholinesterase in complex with donepezil (PDB code: 1EVE). Taken and adapted from Ref. [36].

**Figure 8 pharmaceutics-14-01914-f008:**
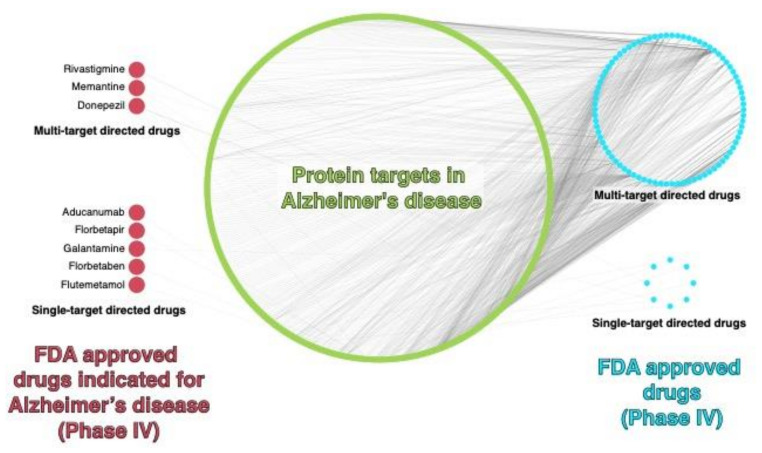
Drug–Protein Interaction network in AD. FDA-approved drugs (phase 4) with reported activity against AD-targets, data were retrieved on July 2022 from ChEMBL database (v31) using the “phembl_value” as search criteria. For further details, see the Appendix A.

**Figure 9 pharmaceutics-14-01914-f009:**
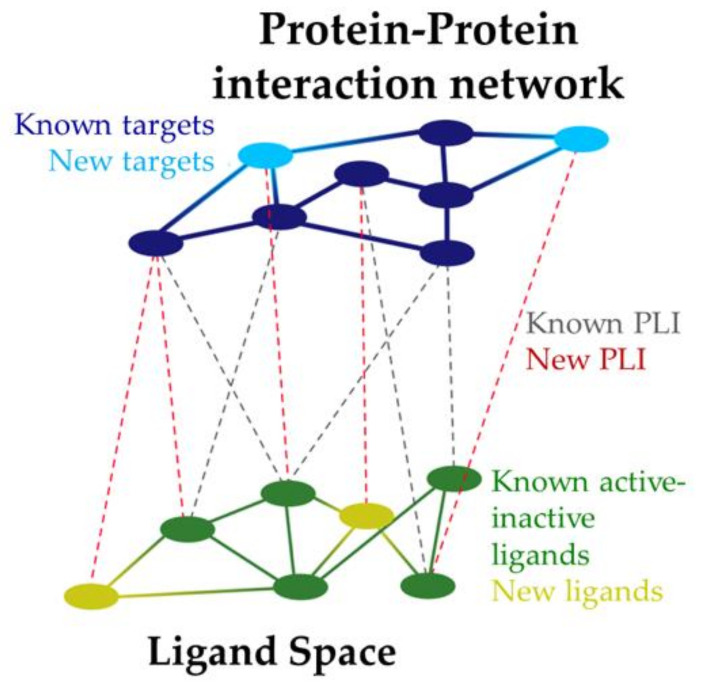
Protein–ligand interaction (PLI) maps in system pharmacology.

**Figure 10 pharmaceutics-14-01914-f010:**
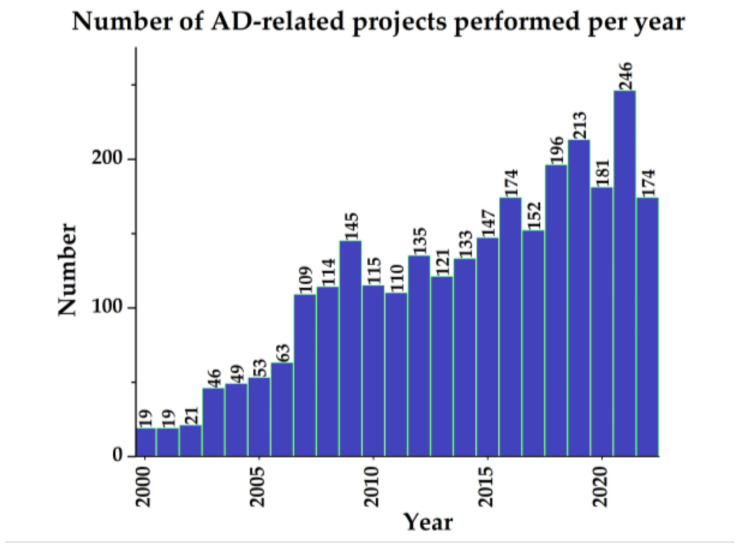
Number of AD-related projects per year. In addition, fewer than 50 projects were carried out in the 1990s, and by 2023 there are already 10 proposed. (Data retrieved on July 2022 from www.clincialtrials.gov).

**Table 1 pharmaceutics-14-01914-t001:** Examples of software and databases developed exclusively for Alzheimer’s disease drug design research.

Software/Platform	Description	Link	Reference
AlzhCPI	With HTML and CSS technology that provides models and important fragments for MTDLs against AD	http://rcidm.org/AlzhCPI	[40]
AlzPlatform	AD-specific chemogenomics database based on ligands	http://www.cbligand.org/AD	[41]
HENA	Heterogeneous network-based dataset for Alzheimer’s disease	https://github.com/esugis/hena	[42]
NIAGADS	National Institute on Aging Genetics of Alzheimer’s Disease Data Storage Site	https://www.niagads.org/	[43]

**Table 2 pharmaceutics-14-01914-t002:** Description and number of treatments against Alzheimer’s disease (Data retrieved on July 2022 from www.clinicaltrials.gov).

Treatment Type	Number of Associated Projects	Description
Drug	1353	Analytical/experimental study. The patient is treated with different drugs. In the cases reported, 105 have used donepezil, 4 rivastigmine, and 4 galantamine, either in the absence of or in addition to other drugs and treatments.
Behavioral	425	Observational study. The patient undergoes therapies, lifestyle changes, sports, and cognitive activities to improve memory. It may or may not be accompanied by other types of therapies. Family therapy and psycho-emotional support are included.
Device	263	Interventional study where devices such as transcranial alternating current stimulation (tACS) and deep brain stimulation (DBS) are used to evaluate possible improvements in patient responses.
Procedure	112	The patient undergoes procedures such as yoga, hypnosis, surgery, or acupuncture.
Dietary supplement	65	New types of diets are implemented for the patient with specific supplements such as vitamin E, curcumin, and omega 3, among others.

**Table 3 pharmaceutics-14-01914-t003:** Examples of MTDLs and their biological activity against targets involved in Alzheimer’s disease.

Compound	Hybrid-Related	Biological Activity IC_50_ (µM)	Reference
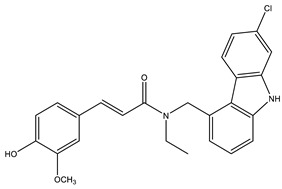	Carbazole-curcumin	AChE: 6.9 ± 0.9 BuChE: 2.8 ± 0.4	[101]
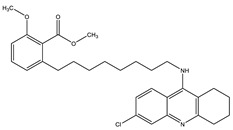	Tacrine–Anacardic acid	AChE: 2.54 ± 0.07 BuChE: 0.265 ± 0.027	[27]
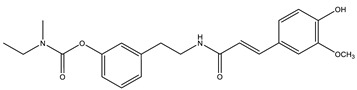	Rivastigmine	AchE at 1 µM = 24.43% BuChe at 1 µM = 72.30% At 10 µM 2.2% of Aβ self aggregation	[102]
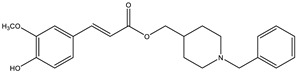	Donepezil–curcumin	AChE: 0.46 BuChE: 24.97	[103]
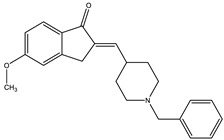	Donepezil	AChE: 0.029 BACE1: 0.33	[104]
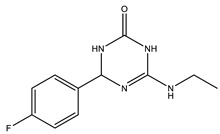	Cyclic amide group	BACE1: 16.0 GSK-3: 7.1	[105]
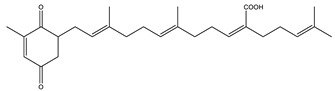	Sargaquinoic-acid	AChE: 69.3 BuChE: 10.5 BACE1: 12.1	[106]

## Data Availability

Not applicable.

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
