# Peer review of "New Drug Design Avenues Targeting Alzheimer’s Disease by Pharmacoinformatics-Aided Tools"

_pharmaceutics, 2022, doi:10.3390/pharmaceutics14091914_

Round 1

Reviewer 1 Report

In my opinion the manuscript entitled “New drug design avenues targeting Alzheimer's disease by pharmacoinformatics-aided tools”, can be accepted after a minor revision.

The manuscript presents a well-documented scientific review on Alzheimer's disease (AD) focusing on computational polypharmacology and pharmacoinformatics tools to address this complex disease. AD being considered a multifactorial disease, the complexity of the problem, has guided research to understanding AD pathology, particularly by targeting several mechanisms and developing multi-target-directed ligands (MTDLs).

The subject of the review is topical, although it is studied quite intensively and there are several review article published in recent years that approach AD from different perspectives, the subject being of interest, there is room for updating the information. As an observation, the reference [6] of the co-authors Ramirez, D. and Arrue, L. addresses part of the information found in the current work.

The “6. Drug Repurposing strategies” section is quite short, it can be expanded with important existing data, for example the study of Kumar S. and collaborators [Kumar S, Chowdhury S, Kumar S. In silico repurposing of antipsychotic drugs for Alzheimer's disease. BMC Neurosci. 2017 Oct 27;18(1):76. doi: 10.1186/s12868-017-0394-8. PMID: 29078760; PMCID: PMC5660441.]

In “Figure 2. FDA approved drugs for the treatment of Alzheimer's disease.” please also include Aducanumab. See the document “FDA-approved treatments for Alzheimer’s” https://www.alz.org/media/documents/fda-approved-treatments-alzheimers-ts.pdf

The number of references is reasonable and most of them are relatively new.

Author Response

Thank you for your review and valuable comments. Following are the responses according to your comments:

Reviewer 1

In my opinion the manuscript entitled “New drug design avenues targeting Alzheimer's disease by pharmacoinformatics-aided tools”, can be accepted after a minor revision.

The manuscript presents a well-documented scientific review on Alzheimer's disease (AD) focusing on computational polypharmacology and pharmacoinformatics tools to address this complex disease. AD being considered a multifactorial disease, the complexity of the problem, has guided research to understanding AD pathology, particularly by targeting several mechanisms and developing multi-target-directed ligands (MTDLs).

The subject of the review is topical, although it is studied quite intensively and there are several review article published in recent years that approach AD from different perspectives, the subject being of interest, there is room for updating the information. As an observation, the reference [6] of the co-authors Ramirez, D. and Arrue, L. addresses part of the information found in the current work.

The “6. Drug Repurposing strategies” section is quite short, it can be expanded with important existing data, for example the study of Kumar S. and collaborators [Kumar S, Chowdhury S, Kumar S. In silico repurposing of antipsychotic drugs for Alzheimer's disease. BMC Neurosci. 2017 Oct 27;18(1):76. doi: 10.1186/s12868-017-0394-8. PMID: 29078760; PMCID: PMC5660441.]

RTA. Thank you for your comment. Section 6 was revised and modified. New references were added together with the recommended paper. The updated version can be found in the section 6 of the manuscript. We also included a network-pharmacology analysis of the current drugs approved and susceptible to be repurposed for AD.

In “Figure 2. FDA approved drugs for the treatment of Alzheimer's disease.” please also include Aducanumab. See the document “FDA-approved treatments for Alzheimer’s” https://www.alz.org/media/documents/fda-approved-treatments-alzheimers-ts.pdf

RTA. The figure showing the FDA-approved drugs for AD was completed by adding the monoclonal antibody aducanumab. The new figure can be found in the manuscript as Figure 3.

The number of references is reasonable and most of them are relatively new.

Reviewer 2 Report

The manuscript written by Arrué et al. is about discussing pharmacoinformatic strategies mainly as a tool to make predictions and facilitate drug design processes in the field of Alzheimer’s disease (AD). Other approaches like drug repurposing strategies have been also included. The manuscript is interesting, well organized, and affords valuable information in the field of drug design. There are a couple of issues that prevent me from accepting this manuscript in the present form. The following suggestions should be included prior to publishing this manuscript in Pharmaceutics:

1. Introduction section. The authors describe the main hypotheses involved in AD. To facilitate the understanding of the disease, I suggest including the main hypotheses on the pathogenesis of AD in a new figure.

2. Page 8; Line 291. Multi-target directed ligands (MTDLs) for AD. Although the authors have included a figure, I think they should also include a new table describing the most relevant examples involving MTDLs. This will help readers explore this approach and easily find other potential structures.

3. Page 11; Line 434. Drug repurposing strategies. I think a new table should be included in this section showing the potential drugs that have been used as novel treatments against AD. In this sense, the authors should also include potential targets these drugs may interact with.

Author Response

Thank you for your review and valuable comments. Following are the responses according to your comments.

Reviewer 2

The manuscript written by Arrué et al. is about discussing pharmacoinformatic strategies mainly as a tool to make predictions and facilitate drug design processes in the field of Alzheimer’s disease (AD). Other approaches like drug repurposing strategies have been also included. The manuscript is interesting, well organized, and affords valuable information in the field of drug design. There are a couple of issues that prevent me from accepting this manuscript in the present form. The following suggestions should be included prior to publishing this manuscript in Pharmaceutics:

  1. Introduction section. The authors describe the main hypotheses involved in AD. To facilitate the understanding of the disease, I suggest including the main hypotheses on the pathogenesis of AD in a new figure.

RTA. Thanks for your comments and suggestions, the new image depicting the pathophysiology of Alzheimer's disease related to the main hypotheses described in this review was added to the manuscript as Figure 1.

  1. Page 8; Line 291. Multi-target directed ligands (MTDLs) for AD. Although the authors have included a figure, I think they should also include a new table describing the most relevant examples involving MTDLs. This will help readers explore this approach and easily find other potential structures.

RTA. Table 3 was added to the manuscript, with some structures and relevant information on MTDLs reported in recent years. Additionally, Figure 5 was incorporated showing the bioactive compounds with activity against Alzheimer's disease targets currently reported in the ChEMBL database v31, single and multitarget directed ligands are showed. Further information was deposited in the Open Science Framework project “New drug design avenues targeting Alzheimer's disease by pharmacoinformatics-aided tools” DOI: 10.17605/OSF.IO/BY86R. Link: https://osf.io/by86r/. 

  1. Page 11; Line 434. Drug repurposing strategies. I think a new table should be included in this section showing the potential drugs that have been used as novel treatments against AD. In this sense, the authors should also include potential targets these drugs may interact with.

RTA: Thanks for your suggestions. We now include a fully detailed drug-protein interaction network (figure 8 – Table S1) were FDA-approved drugs (phase 4) with reported activity against AD-targets are displayed. Data was retrieved from ChEMBL database (v31) using the “phembl value” as search criteria.  The Table S1 contains all AD-related targets and drugs with bioactive activity (pchembl value) reported.

Reviewer 3 Report

Lily et al. reviewed the recent drug design approaches targeting AD, highlighting the role of polypharmacology and pharmacoinformatics against complex diseases like AD. While the research appeared technically sound, I suggest several revisions should be considered as follows:

1. The methods listed in the reviews are typical computer-aided drug design tactics, including multitarget directed ligands, virtual screening, molecular dynamics, pharmacophore modeling, and drug repurposing, and then the authors gave examples to illustrate their specific application in AD. To make the review clearer, the authors should briefly specify the fundamentals for these methods in the introduction part. A table containing every strategy, its aim, and the website or software to apply this strategy should be very helpful. More importantly, the authors need to emphasize which of these methods are particularly effective or unique for AD.

2. In section 1.1.2, as for the amyloid hypothesis of AD, several important studies about Aβ56 currently cause controversy. The authors should add this to update the status of the amyloid hypothesis.

3. In line 176, the artificial neural network could also be unsupervised. The authors should stress this concept in terms of the labels of the dataset. ML methods are highly data-driven. Are there any high-quality datasets about AD? This would be of high review value and can be added in section 5.2. 

Author Response

Thank you for your review and valuable comments. Following are the responses according to your comments:

Reviewer  3

Lily et al. reviewed the recent drug design approaches targeting AD, highlighting the role of polypharmacology and pharmacoinformatics against complex diseases like AD. While the research appeared technically sound, I suggest several revisions should be considered as follows:

  1. The methods listed in the reviews are typical computer-aided drug design tactics, including multitarget directed ligands, virtual screening, molecular dynamics, pharmacophore modeling, and drug repurposing, and then the authors gave examples to illustrate their specific application in AD. To make the review clearer, the authors should briefly specify the fundamentals for these methods in the introduction part. A table containing every strategy, its aim, and the website or software to apply this strategy should be very helpful. More importantly, the authors need to emphasize which of these methods are particularly effective or unique for AD.

RTA. Thanks for the comments. Due to the great variety of literature referring to this aspect, an introductory paragraph has been added to the manuscript (lines 153 – 185) where different papers and reviews are referenced with the requested information. This is because the methods and software used in drug design are large and their application will depend on the objectives and interests of the researcher, so summarizing a table with the information might not be enough to cover the request in a complete manner. Moreover, Table 1 has been incorporated to shows some examples of software and databases used only for Alzheimer's disease studies.

  1. In section 1.1.2, as for the amyloid hypothesis of AD, several important studies about Aβ56 currently cause controversy. The authors should add this to update the status of the amyloid hypothesis.

RTA. Section 1.1.2 was revised and modified. New references were added to describe the status and controversy associated with the amyloid hypothesis. The modification made can be found in the corresponding section of the updated manuscript.

  1. In line 176, the artificial neural network could also be unsupervised. The authors should stress this concept in terms of the labels of the dataset. ML methods are highly data-driven. Are there any high-quality datasets about AD? This would be of high review value and can be added in section 5.2. 

RTA. Thanks for your comments and suggestions. The changes were incorporated in line 497 and  section 5.2 as suggested.

Round 2

Reviewer 2 Report

The authors have addressed all the reviewer's suggestions and I recommend the publication of this manuscript in Pharmaceutics

Reviewer 3 Report

This revised manuscript can be accepted for publication.